# Acrolein: A Potential Mediator of Oxidative Damage in Diabetic Retinopathy

**DOI:** 10.3390/biom10111579

**Published:** 2020-11-20

**Authors:** Moaddey Alfarhan, Eissa Jafari, S. Priya Narayanan

**Affiliations:** 1Clinical and Experimental Therapeutics, College of Pharmacy, University of Georgia and Charlie Norwood VA Medical Center, Augusta, GA 30912, USA; malfarhan@augusta.edu (M.A.); ejafari@augusta.edu (E.J.); 2Vision Discovery Institute, Augusta University, Augusta, GA 30912, USA

**Keywords:** diabetic retinopathy, oxidative stress, acrolein, vision, polyamine oxidation

## Abstract

Diabetic retinopathy (DR) is the leading cause of vision loss among working-age adults. Extensive evidences have documented that oxidative stress mediates a critical role in the pathogenesis of DR. Acrolein, a product of polyamines oxidation and lipid peroxidation, has been demonstrated to be involved in the pathogenesis of various human diseases. Acrolein’s harmful effects are mediated through multiple mechanisms, including DNA damage, inflammation, ROS formation, protein adduction, membrane disruption, endoplasmic reticulum stress, and mitochondrial dysfunction. Recent investigations have reported the involvement of acrolein in the pathogenesis of DR. These studies have shown a detrimental effect of acrolein on the retinal neurovascular unit under diabetic conditions. The current review summarizes the existing literature on the sources of acrolein, the impact of acrolein in the generation of oxidative damage in the diabetic retina, and the mechanisms of acrolein action in the pathogenesis of DR. The possible therapeutic interventions such as the use of polyamine oxidase inhibitors, agents with antioxidant properties, and acrolein scavengers to reduce acrolein toxicity are also discussed.

## 1. Introduction

Diabetic retinopathy (DR), a complication of diabetes, is a significant public health issue and the leading cause of vision loss among working-age adults [1]. DR accounts for almost 40% of diabetes mellitus complications in patients aged ≥40 years [2]. It is estimated that the prevalence of any form of DR is about 93 million worldwide [3]. DR patients’ quality of life is negatively impacted, affecting their mental health and general vision, and is decreased with the duration of the disease [4]. DR develops in both type 1 and type 2 diabetic patients. It is estimated that individual lifetime risk of DR is up to 90% in patients with type 1 diabetes mellitus and 50–60% in those with type 2 diabetes mellitus [5,6]. The unavailability of effective treatments to prevent the incidence or progression of DR is a major clinical problem. The current therapeutic avenues for DR are anti-VEGF agents, anti-inflammatory drugs, and laser treatment. However, they treat advanced stages of the disease, particularly the vascular damage, and have adverse side effects. Hence there is a great need for the identification of novel therapeutic interventions for DR.

All the components of the retinal neurovascular unit are disrupted by diabetes [7]. The precise mechanisms through which diabetes causes retinal damage and degeneration remain to be fully established. Extensive evidences have documented that oxidative stress plays a critical role in the pathogenesis of DR. Formation of reactive oxygen species (ROS) has been shown to play important roles in DR progression [8]. Several pathways and molecules have been studied to be associated with elevated oxidative stress in the diabetic retina [8,9,10,11,12]. Dysregulation in the polyamine pathway is involved in the pathogenesis of diabetes and diabetic retinopathy [13,14,15]. The oxidative products of polyamine metabolism can induce cellular damage and cell death [16,17,18]. Reactive aldehydes and hydrogen peroxide (H_2_O_2_) are generated from the metabolism of polyamines [19]. These reactive aldehydes are spontaneously converted to acrolein [16,20], a potent mediator of oxidative damage in various pathologies [21,22].

Acrolein is involved in different pathologies and plays a vital role in the progression of atherosclerosis [23], cardiovascular disease [24], pulmonary inflammation [25], and kidney disease [26]. Additionally, acrolein is involved in neurodegenerative diseases such as Alzheimer’s disease [27], multiple sclerosis [28], brain infarction [29,30,31,32], and spinal cord injury [33]. Acrolein exposure may cause cognitive problems and may lead to serious neurocognitive effects on humans [34]. Acrolein’s harmful effects are mediated through multiple mechanisms, including DNA damage, inflammation, ROS formation, protein adduction, membrane disruption, endoplasmic reticulum stress, and mitochondrial dysfunction [21]. In vitro studies revealed that injury induced by acrolein was more toxic than ROS [20,35] and other aldehydes produced during polyamine metabolism [36]. This oxidative damage caused by acrolein leads to mitochondrial dysfunction, membrane disruption, and increased apoptosis [21,37,38]. Recently, several investigations have reported the emerging role of acrolein in the progression of DR. The current review summarizes the existing literature on the various sources of acrolein, the impact of acrolein in the generation of oxidative damage in the diabetic retina, and the mechanisms of acrolein action in the pathogenesis of DR.

## 2. Acrolein: Sources and Metabolism

Acrolein is a highly reactive unsaturated aldehyde. It is found as a contaminant in food, air, and water. The major sources of acrolein include dietary, environmental, and endogenous sources. Acrolein is an environmental pollutant generated in the environment during incomplete combustion of plastic, combustion of petrochemical fuels, and tobacco smoke [39]. In addition, acrolein is produced from overheated vegetable oils and animal fats [40]. Endogenously, acrolein is formed during the peroxidation of polyunsaturated fatty acids [41] or polyamine catabolism [16]. Free radicals generated from ROS react with the polyunsaturated fatty acids (PUFAs), forming lipid radicals and lipid peroxyl radicals. These are highly unstable and are further degraded into reactive, secondary products such as acrolein, 4-hydroxynonenal (4-HNE), and malondialdehyde (MDA) [42,43]. These aldehydes react with cellular proteins, leading to advanced lipoxidation end products (ALEs) formation [42,43,44]. The other endogenous source for acrolein formation is the oxidation of spermine and spermidine [39]. During polyamine oxidation, H_2_O_2_ and aldehydes, including 3-acetamidopropanal (3-AAP), amino aldehydes 3-aminopropanal (3-AP), and 4-aminobutanal (4-AB), are formed [36,45]. Spermine and spermidine are converted to spermidine and putrescine, respectively, through back-conversion by SSAT (spermidine/spermine N1-acetyltransferase) to N1-acetylated polyamines. N1-acetylated polyamines are oxidized by polyamine oxidase (PAO) to produce spermidine and putrescine, respectively [46]. In addition to the PAO oxidation pathway, spermine can also be oxidized by spermine oxidase (SMOX) to spermidine directly [46]. Therefore, both PAO and SMOX oxidation pathways are involved in generating acrolein from polyamines [47]. The formation of acrolein through lipid peroxidation and polyamine oxidation pathways are presented in Figure 1A,B.

Acrolein is highly soluble in water and alcohol and can cross the membranes by passive diffusion. The mode of elimination of acrolein is by conjugation with glutathione (GSH) in the liver and *N*-acetylation of the resultant cysteine conjugate to form *S*-(3-oxopropyl)-*N*-acetylcysteine (OPMA) in the kidney. OPMA is reduced to form *N*-acetyl-*S*-(3-hydroxypropyl)-l-cysteine (or 3-hydroxypropyl mercapturic acid, 3HPMA), the major urinary metabolite of acrolein, and an oxidative route which yields *N*-acetyl-*S*-[2-carboxyethyl]-l-cysteine (or 2-carboxyethyl-mercapturic acid, CEMA), a minor urinary metabolite [39,48,49]. These metabolites, 3HPMA and CEMA offer considerable promise as urinary biomarkers. Oxidation of acrolein can lead to the formation of acrylic acid. Enzyme-mediated epoxidation of acrolein produces glycidaldehyde, which can react with water to yield glyceraldehyde or form a conjugate with GSH. Additional acrolein metabolites detected in urine include 3-hydroxy propionic acid, malonic acid, and *N*-acetyl-*S*-(2-carboxy-2-hydroxyethyl)-cysteine [39]. Since acrolein is highly unstable, measurements of acrolein metabolites are useful indicators of its internal levels. Various methods are implemented for the detection of acrolein conjugates or metabolites. Liquid chromatography-tandem mass spectrometry (LC/MS) analysis and high-performance liquid chromatography (HPLC) are widely used for the detection of acrolein metabolites in biological fluids [50,51,52,53]. The presence of acrolein conjugates is studied by slot blot assays [51], ELISA [54,55], or immunochemical methods [56,57].

## 3. Acrolein in the Progression of Diabetic Retinopathy

### 3.1. Acrolein and Diabetes

Studies have reported the association of acrolein in both type 1 and type 2 diabetes. Elevated urinary level of conjugated acrolein is reported in patients with type I and type 2 diabetes [58,59]. These clinical studies illustrated that elevated urinary acrolein level is linked to the diabetic stage and significantly correlated to the glycemic status [58,59]. In a group of younger type 1 diabetic patient, the urine level of conjugated acrolein, along with other markers of oxidative stress (pentosidine, pyrraline, and 8-hydroxy-2′-deoxyguanosine), was shown to be significantly elevated than healthy subjects [58]. The study performed by Daimon et al. and colleagues [59] in type 2 diabetic patients showed significantly increased levels of acrolein adduct, pentosidine, and pyrraline in urine samples. Furthermore, glycemic control parameters, such as fasting plasma glucose and HbA1c (hemoglobin A1c, the glycated hemoglobin level), were observed to be correlated with the urinary levels of these markers. The acrolein adduct level was higher in subjects with smoking habits than in those without the habit in the diabetic and nondiabetic groups [59]. In addition, acrolein protein adducts are associated with diabetes-related complications such as diabetic nephropathy [60] and diabetic retinopathy [61].

Acrolein feeding elicited dyslipidemia in a mouse model, with increased circulating cholesterol and triglycerides [62,63]. The mechanism behind this acrolein-induced dyslipidemia involved the processes of lipid synthesis and clearance. Acrolein treatment reduced VLDL clearance via decreasing hepatic lipase activity and downregulating LDL receptors, leading to modulating the VLDL clearance and increasing the circulating cholesterol and triglycerides [62]. Acute acrolein feeding modified the expression of plasma and hepatic genes involved with lipid synthesis and trafficking and increased plasma triglycerides [63]. Diabetes is well known to cause dyslipidemia (a condition with abnormal cholesterol levels and lipid in the blood), which increases lipid peroxidation even in well-controlled diabetic patients [64]. Dyslipidemia and lipid peroxidation can lead to increased acrolein production, thus causing diabetes-related complications [41,64]. In addition to lipid peroxidation, increased activity of polyamine oxidases can also cause acrolein formation. An earlier report has indicated a positive correlation between hyperglycemia and PAO activity and that the activity of PAO is positively correlated to the level of HbA1c [65]. These observations suggest that diabetic complications can possibly be triggered by acrolein produced either via lipid peroxidation or polyamine oxidation. A recent report indicated an age-dependent increase in acrolein conjugate, polyamine oxidation, and SMOX in the liver of human subjects [66].

### 3.2. Pathogenesis of DR

DR’s incidence and progression are affected by various factors, including duration of diabetes, age, pregnancy, hyperglycemia, hyperlipidemia, obesity, smoking, etc. [10]. It is recognized as a neurovascular disease, and neurodegeneration and vasculopathy are the major hallmarks of DR progression. Diabetes-induced retinal neurodegeneration is characterized by the progressive thinning of the nerve fiber layer, loss of retinal ganglion cells, and alterations in visual function [15,67,68,69,70,71]. While neurodegeneration is considered as an early event, the clinical classification of DR is based on its vascular complications. These include increased vascular permeability, endothelial dysfunction, vascular degeneration, and pathological angiogenesis [72,73]. In addition to neuronal and vascular changes, the diabetic retina exhibits increased inflammatory changes [74,75,76] and glial activation [77,78]. Müller cells, the principal glia of the retina, serve to maintain the extracellular environment’s homeostasis and play an important role in the pathogenesis of DR [79]. In addition, Müller cells are a part of the inner blood–retinal barrier cells and regulate retinal blood flow [79]. It is known that Müller cells become activated in the retina during diabetes, and the expression of the glial fibrillary acidic protein (GFAP) is increased in the Müller cells and is a characteristic feature of the diabetic retina [80]. Several mechanisms/pathways have been demonstrated to be involved in the pathogenesis of DR [81,82]. Oxidative stress has been established as a significant contributor to DR pathogenesis. It has been confirmed that metabolic pathways, including advanced glycation end products (AGEs), polyol, hexosamine, and protein kinase C (PKC), are activated under diabetic conditions. Accumulation of by-products of these pathways can induce oxidative stress through the formation of ROS and nitrogen-oxygen species. Studies from our laboratory are the first to demonstrate polyamine oxidation in mediating neuronal and vascular damage in the retina [83,84]. Studies from our group [15,83] and others [12,85] have shown the involvement of SMOX, a crucial enzyme in the polyamine oxidation pathway in DR. Altered levels of polyamines are reported in the vitreous samples from patients with proliferative DR [86]. Endogenous spermine, a polyamine, demonstrated role in the distribution and regulation of voltage-dependent calcium channels in the diabetic retina [87].

### 3.3. Acrolein Conjugates in the Diabetic Retina

Studies have shown the upregulation of lipid peroxidation and its metabolites in the serum of DR patients [88,89] as well as in the retinas of experimental diabetic models [90,91], emphasizing the role of lipid peroxidation in the progression of diabetes. Several investigations have demonstrated the involvement of polyamine oxidation and lipid peroxidation as the major pathways of acrolein generation in the diabetic retina [15,92,93]. Diabetic patients with DR are found to have much more lipid peroxidation products compared with those without DR [94]. Acrolein is a strong electrophile and hence shows high reactivity with proteins, DNA, and RNA. Acrolein reacts with the sulfhydryl group of cysteine, the imidazole group of histidine, and the amino group of lysine, to form mainly Michael addition-type adducts or Schiff base cross-links [95,96]. Protein adduction by acrolein can cause significant protein modifications leading to alterations in protein functions. FDP-lysine, also known as a biomarker of acrolein, is formed when acrolein conjugates with lysine residue to form *N*^ε^-(3-formyl-3,4-dehydropiperidino)lysine (FDP-lysine) [41]. An earlier study consisting of type 1 and type 2 diabetic patients investigated the correlation between the level of FDP-lysine, (serum and hemoglobin), and the severity of diabetic retinopathy [97]. Compared with the control group, the serum and hemoglobin levels of FDP-lysine were significantly increased in diabetic patients. The levels of hemoglobin FDP-lysine were increased in patients with proliferative retinopathy compared with patients without retinopathy and with nonproliferative retinopathy, however, no association was observed between serum FDP-lysine and severity of DR. FDP-lysine is increased significantly in the vitreous fluids of patients with proliferative diabetic retinopathy (PDR) [98]. Pieces of evidence demonstrated increased levels of FDP-lysine or conjugated acrolein in the retina of diabetic rodents [15,78,99]. In an earlier study, significantly elevated accumulation of FDP-lysine in the Müller cells was observed in the retinas of diabetic animals, and the accumulation was shown to be progressive with the duration of diabetes [99]. Along with other oxidative stress markers, acrolein was reported to be increased in the retinas of diabetic rats [100,101,102]. Altogether, these observations suggest a correlation between acrolein formation and the progression of DR.

Aldehyde dehydrogenases (ALDH) are the enzymes that metabolize aldehydes to lesser toxic compounds [103]. ALDH1A1 is one of the ALDH enzymes that participate in lipid peroxidation detoxifying aldehydes [104,105]. Interestingly, retinas of diabetic rats showed a reduced transcription and expression of ALDH1A1 and reduced activity of ALDH [57]. Acrolein is detoxified when it is conjugated with glutathione (GSH), and this reaction is catalyzed by glutathione S transferase (GST) [39]. However, a reduction in the GSH level was observed in diabetic rat retinas [106]. In an in vitro experimental model, retinal pigment epithelial (RPE) cells showed a reduction in GSH level and a reduction in GST activity when cells were treated with acrolein [107]. These studies suggest the impact of acrolein in elevating oxidative stress in diabetes-induced pathologies in the retina.

## 4. Mechanism of Action of Acrolein in the Diabetic Retina

Acrolein mediates its function by diverse mechanisms. These include direct mechanisms of acrolein toxicity such as protein and DNA adduction and via indirect mechanisms, including oxidative, mitochondrial, and ER stress [21]. Protein modifications by protein adduction is a major mechanism of acrolein-mediated cellular dysfunction. Both in vitro and in vivo evidence have shown that acrolein causes oxidative damage. In the following section, the central mechanisms of acrolein-induced damages investigated in the diabetic retina are summarized.

### 4.1. Depletion of Antioxidants

The compromised antioxidant defense system is one of the causes of elevated oxidative stress in the diabetic retina. This is a major mechanism by which acrolein induces cellular oxidative stress and is executed by the formation of acrolein protein adducts. Because of acrolein’s significant reactivity with thiols, GSH is one of the primary targets of acrolein-mediated injury [39]. Even though the reaction of GSH with acrolein is essential for the endogenous removal of acrolein, depletion of GSH reserves reduces the ability to handle additional oxidative stress [108]. Acrolein decreased the levels of antioxidants such as GSH, intracellular glutathione peroxidase (G-Px), GST, and superoxide dismutase (SOD) in RPE cells [109,110,111]. Acrolein exposure also reduced the total antioxidant capacity (T-AOC) and the expression of glutamate-cysteine ligase (GCL), the enzyme that controls GSH production [111]. Moreover, RPE cells exposed to acrolein showed a reduction in nuclear factor-e2-related factor 2 (Nrf2), a regulator of antioxidant response [109,110,111]. GSH expression was shown to be decreased in the rat retinal Müller cell line TR-MUL5 exposed to acrolein treatments [112]. Studies have shown that acrolein depletes the cellular antioxidant levels (including GSH) by conjugating with thiol groups [113,114]. In an experimental model of diabetes, the accumulation of FDP-lysine in Müller cells was increased and associated with an elevated level of heme oxygenase-1 (HO-1) [78], a marker of oxidative stress [115].

### 4.2. Protein Carbonyl Formation

The formation of protein carbonyls is a major cause of oxidative stress in the diabetic retina. Acrolein can carbonylate proteins to generate reactive carbonyl species [110,116] and increase the oxidative damage by protein carbonylation [96,117]. Recent reports show that protein carbonyl levels are elevated in DR patients [118,119,120] and experimental models [121,122]. The presence of protein carbonyls is also reported in models of photoreceptor degeneration [123] and glaucoma [124]. Acrolein exposure demonstrated to increase the generation of ROS in RPE cells [107,109,110,111] and a rat retinal Müller cell line [112]. The increased oxidative stress in RPE cells caused DNA fragmentation and protein oxidation by increasing protein carbonyl levels [109,110,111]. However, the specific mechanisms by which protein carbonyls are increased in DR needs further investigation.

### 4.3. Mitochondrial Dysfunction

Oxidative damage-mediated mitochondrial dysfunction is a major mechanism for oxidative stress in DR [125]. While mitochondria are essential for cell growth and energy production, they are also responsible for ROS formation [126]. Acrolein is known as a mitochondrial toxin and can impair mitochondrial respiratory function [38,127]. In RPE cells, acrolein induced mitochondrial dysfunction by decreasing mitochondrial membrane potential and mitochondrial viability and by increasing the intracellular Ca^2+^ [107,109,110,111]. Additionally, acrolein exposure reduced oxygen consumption, mitochondrial complexes (I, II, and V) activities, and ATP content in RPE cells [107,109,110,111]. Acrolein-induced mitochondrial dysfunction is demonstrated in the rat brain [128,129]. Carbonylation of mitochondrial proteins by acrolein has been shown as a mechanism of neuronal death studied using neuron-like PC12 cells [116].

## 5. Impact of Acrolein on DR Pathogenesis

It is well known that diabetes trigger cellular changes in all component of the retinal neurovascular unit, including neurons, glial cells, and blood vessels [7]. The diabetic retina is characterized by vascular changes, glial activation, and neuronal loss. As illustrated in this review, acrolein is involved in DR pathogenesis and its progressive nature. Possibly, acrolein accumulation might affect each element of the neurovascular retina. The potential mechanisms of acrolein-induced damage in the diabetic retina are presented in Figure 2.

### 5.1. Inflammation

Inflammation plays a significant role in DR pathogenesis. FDP-lysine accumulation in the diabetic rats was associated with an increase in Müller cells gliosis, upregulation of receptor for advanced glycation end products (RAGE), and calcium-binding protein B (S100B), resulting in the microglial activation and the secretion of inflammatory mediators such as CCL2, IL-1b, and ICAM-1 [78]. Furthermore, in vitro studies demonstrated that the increase in oxidative stress by acrolein exposure increased the protein expression and mRNA level of inflammatory chemokine CXCL1 in rat retinal Müller cell line [112]. The incubation of human Müller cells (MIO-M1) with FDP-lysine human serum albumin increased the mRNA expression of inflammatory molecules, interleukin-6 (IL-6), and tumor necrosis factor-α (TNFα) [99].

### 5.2. Neurodegeneration

Diabetes deteriorates the visual function, and electroretinogram studies can assess the functional changes. An in vivo analysis illustrated that after 7 weeks of diabetes, rats experienced a reduction in a- and b-waves and increased the summed oscillatory potential [78]. An elevation in the FDP-lysine level accompanied these alterations in diabetes-induced retinal function. These results might indicate a crucial role of acrolein in the diabetes-induced alterations in visual function. Studies have shown that treatment with 2-hydrazino-4,6-dimethylpyrimidine, an acrolein scavenger, and MDL72527 (N, N′-Bis(2,3-butadienyl)-1,4-butanediamine dihydrochloride), a PAO/SMOX inhibitor, improved the visual function in diabetic animals [15,78]. Immunofluorescence evidence from our laboratory has shown an increase in conjugated acrolein levels in the ganglion cell layer and inner nuclear layer in STZ diabetic mice [15]. This increase was associated with a reduction in the retinal function and retinal thinning [15]. These observations support that acrolein is involved in the neurodegenerative process in the diabetic retina. Treatment with acrolein showed a toxic effect on rat retinal ganglion cell line and was more harmful than other aldehydes produced during polyamine metabolism [36].

### 5.3. Vascular Damage and Blood–Retinal Barrier Integrity

One of the most critical components of the neurovascular unit is the vascular compartment. Dong et al. reported that FDP-lysine is accumulated in endothelial cells and pericytes of PDR patients [130]. The accumulation of FDP-lysine was associated with increased vascular density in patients with PDR [130]. To further confirm the involvement of FDP-lysine in the vascular changes, human retinal microvascular endothelial cells (HRMECs) were treated with a sublethal dose of acrolein, and an increase in cell proliferation and the expression of HO-1 were observed [130]. Incubation with a high dose of acrolein decreased the cell viability of HRMECs [130]. Studies performed by Murata et al. showed that FDP-lysine levels in the endothelial cells of fibrovascular tissues from PDR patients and that treatment of rat retinal capillary endothelial cell line (TR-iBRB2) with acrolein reduced GSH levels in a dose-dependent manner and caused cellular toxicity [13]. RPE cells are involved in the pathogenesis of DR as they maintain the external blood–retinal barrier and produce VEGF and transforming growth factor β (TGFβ). Both VEGF and TGFβ, especially VEGF, plays a vital role in the development of diabetic retinopathy [131,132]. Treatment of ARPE-19 cells with acrolein upregulated VEGF, and TGFβ signaling pathway and these effects further increased when cells were cultured in high glucose medium [61]. Acrolein exposure reduced the viability of RPE cells and mitochondrial membrane potential [109,110,111].

### 5.4. Müller Glial Dysfunction

Studies using experimental models of DR have shown that FDP-lysine is accumulated in retinal Müller cells of diabetic animals [78,99]. The increase in FDP-lysine was associated with Müller glial dysfunction [78]. The upregulation of FDP-lysine in Müller cells could be because they are the primary polyamine storage in the retina [133]. Cell culture studies showed that small doses of acrolein slightly increased the viability TR-MUL5 cells, while the higher doses of acrolein reduced the viability significantly [112]. Acrolein exposure increased the mRNA and the protein expression of inflammatory chemokine CXCL1 in TR-MUL5 cells [112]. Additionally, the increase in CXCL1 by acrolein treatment induced cell migration of TR-MUL5 cells [112]. An earlier study has shown that incubation of human serum albumin (HSA)-conjugated FDP-lysine increased the apoptosis of MIO-M1 cells [99]. It also upregulated VEGF transcription [99], which plays an important role in the angiogenesis and development of proliferative DR [131]. Based on these observations on the impact of acrolein on the components of the blood–retinal barrier (BRB), it is suggested to play a curial role in regulating vascular permeability and neovascularization.

Figure 3 summarizes the molecular targets of acrolein in the development of DR.

## 6. Strategies to Reduce Acrolein Toxicity

Current treatments for DR focus on the late stage of the disease and have side-effects. Based on the increasing evidence on acrolein’s implication in DR pathogenesis, strategies to eliminate or reduce its toxicity are of very high importance. The use of agents with antioxidant properties, SMOX/PAO inhibitors, and acrolein-scavenging agents may offer potential intervention strategies. MDL 72527 is an irreversible inhibitor for polyamine oxidases [134]. The inhibition of SMOX/PAO function with MDL 72527 treatment in diabetic mice decreased the presence of conjugated acrolein in the retina [15]. This reduction by MDL 72527 treatment was associated with improved neuronal survival and retinal function in diabetic mice [15].

Several classes of natural products/synthetic compounds have shown to possess potential as acrolein-trapping agents. These include sulfur (thiol)-containing compounds, a group of nitrogen-containing compounds, and the naturally occurring phenolic compounds with acrolein-scavenging properties [135]. Thiol containing acrolein scavengers are compounds such as GSH, l-cysteine, 2-mercaptoethane sulfonate (MESNA), 2,6-Dithiopurine, and lipoic acid. Pretreatment with alpha-lipoic acid (LA), hydroxytyrosol (HTS), and α-tocopherol protected against the reduction in cell viability caused by acrolein in RPE cells [109,110,111]. These compounds also reduced the mitochondrial dysfunction caused by acrolein exposure [109,110,111]. Furthermore, pretreatment with LA or HTS inhibited the acrolein-induced increase in intracellular Ca^2+^ levels and protected against acrolein-induced oxidative damage by reducing ROS and increasing the cellular antioxidants in RPE cells [109,110,111]. The increased expression of CXCL1 induced by acrolein TR-MUL5 cells was prevented by N-acetylcysteine (NAC) [112]. Many nitrogen (amino)-containing compounds are being used as acrolein scavengers. These include hydralazine, carnosine, aminoguanidine, pyridoxamine, edaravone, and glycyl-histidyl-lysine. A potent acrolein scavenger, 2-hydrazino-4,6-dimethylpyrimidine (2-HDP), significantly decreases the presence of FDP-lysine in the retina of diabetic rats [78]. Moreover, 2-HDP also reduced the activation of microglia and inflammation markers in the retina of diabetic rats. Furthermore, treatment with 2-HDP also attenuated Müller cells gliosis, oxidative stress, distribution of potassium channels, and visual function in the diabetic rats [78]. While acrolein scavengers are still emerging in the field of vision, they are widely investigated in other central nervous system disorders. Treatment with hydralazine, an acrolein scavenger, and an antihypertensive drug demonstrated reduced neuroinflammation and neuropathic pain in a rat model of diabetes [136]. Hydralazine improved behavioral outcome and reduced myelin damage in an experimental model of multiple sclerosis [50,137] and reduced the neural apoptosis, brain edema, and neurological functional deficits in an experimental model of intracerebral hemorrhage [138]. Phenelzine, an FDA-approved monoamine oxidase inhibitor with acrolein-scavenging properties, showed neuroprotection in the experimental models of traumatic brain injury [139,140], multiple sclerosis [141], and spinal cord injury [142]. There are naturally occurring polyphenols with acrolein-scavenging properties. These include epicatechin, cyanomaclurin, theaflavin, phloretin, and phloridzin [135]. However, preclinical studies are yet to be conducted to demonstrate their acrolein-detoxifying properties. Further studies focused on the impact of acrolein-scavenging agents are needed in models of diabetic retinopathy. While defense against oxidative stress is thought to be a major event involved in the inhibition of acrolein’s harmful effects, compounds with both antioxidant activity and acrolein-scavenging capacity should be more impactful.

## 7. Conclusions and Future Prospects

Diabetic retinopathy is a major complication of diabetes, and the unavailability of effective treatment strategies to reduce DR progression is a major problem. Recent studies have shown an emerging role of acrolein, an unsaturated aldehyde and a mediator of oxidative damage, in DR pathogenesis. Polyamine oxidation and lipid peroxidation are the major pathways of acrolein formation in the diabetic retina. Acrolein elevates oxidative stress and mediates cellular damage and dysfunction by mechanisms such as the depletion of antioxidants, formation of protein carbonyls, and mitochondrial dysfunction. Acrolein is shown to be involved in diabetes-induced alterations in the neurovascular retina, such as neurodegeneration, glial dysfunction, vascular injury, and inflammation.

Further investigations are needed to understand better the specific mechanisms of acrolein-induced changes in the diabetic retina. Pharmacological treatment that neutralizes acrolein or inhibits acrolein formation may serve as promising therapeutic options for DR treatment. A comprehensive understanding of the mechanisms involved in acrolein-induced retinal damage in the diabetic retina may help establish acrolein scavenging as a novel therapeutic intervention for diabetic retinopathy.

## Figures and Tables

**Figure 1 biomolecules-10-01579-f001:**
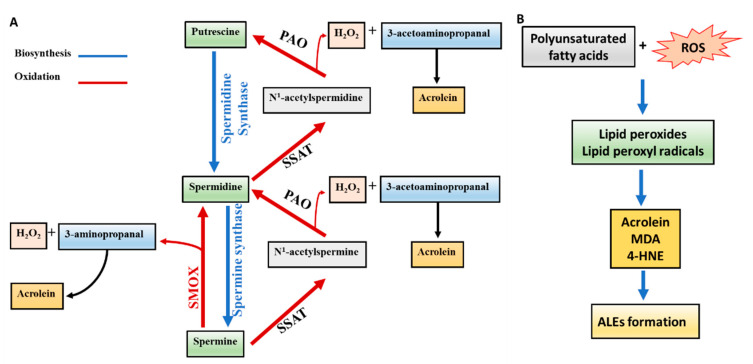
Diagrammatic representation of acrolein formation through polyamine oxidation pathway (**A**) and lipid peroxidation pathway (**B**). PAO: polyamine oxidase; SMOX: spermine oxidase; SSAT: spermidine/spermine N1-acetyltransferase; ALEs: advanced lipid peroxidation end products; MDA: malondialdehyde; ROS: reactive oxygen species; 4-HNE: 4-Hydroxynonenal.

**Figure 2 biomolecules-10-01579-f002:**
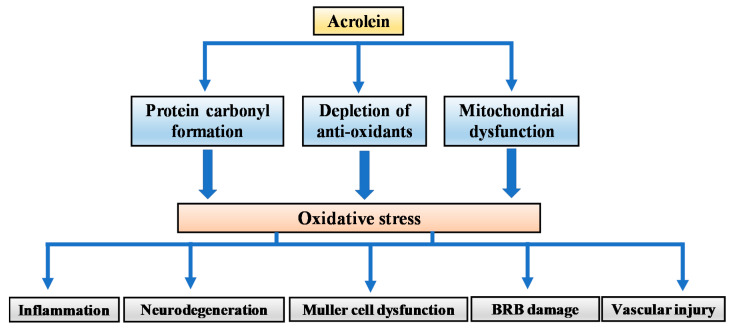
Schematic representation of the potential mechanisms of action of acrolein in the diabetic retina.

**Figure 3 biomolecules-10-01579-f003:**
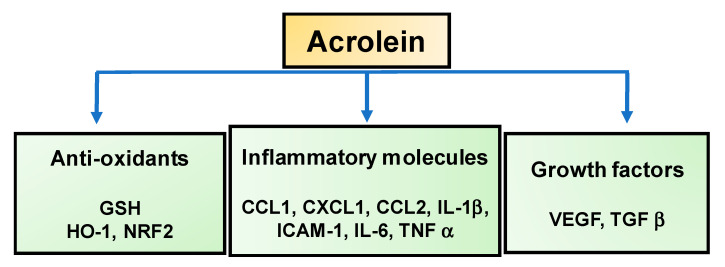
The molecular targets of acrolein in diabetic retinopathy (DR) development. GSH, glutathione; HO-1, heme oxygenase-1, Nrf2, nuclear factor erythroid 2-related factor 2; CCL1, chemokine (C-C motif) ligand 1; CXCL1, C-X-C motif chemokine ligand 1; CCL2, chemokine (C-C motif) ligand 2; ICAM-1, intercellular adhesion molecule 1; IL-6, interleukin 6; IL-1β, interleukin 1 beta, TNF-α, tumor necrosis factor alpha; VEGF, vascular endothelial growth factor and TGF-β, transforming growth factor beta.

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
