# Peer review of "Acrolein: A Potential Mediator of Oxidative Damage in Diabetic Retinopathy"

_biomolecules, 2020, doi:10.3390/biom10111579_

Round 1

Reviewer 1 Report

The review is an elaborate compilation of related literature, however, as the title suggests, there are very limited study that presents a direct link of Acrolein in the development of diabetic retinopathy. Major portion of review is focused on either detailing about Acrolein induced toxicity or the role of biological factors involved in the pathogenesis of DR, but have only limited direct evidences. Author should remove extra text, which seems to be diverting the focus of the manuscript. Also, rather than just including a concluding statement of the referenced articles in the review, author should explain and discuss the statements.

Minor comments

Fig 1 is not necessary. It doesn’t seem to add anything in the manuscript.

In the statement, “The positive correlation between diabetes and acrolein levels in diabetic patients possibly mediated through either lipid peroxidation or polyamines oxidation or a combination of both”, cell type is not clear. Explain.

In the statement “Acrolein feeding increased the circulating levels of cholesterol and triglycerides within 24h in mice”, how acrolein involvement regulates cholesterol and triglycerides is not clear. Explain.

Contribution of Acrolein in type 1 and 2 diabetes is not distinguishable. Discuss with specifications in the review whether the study focused on type 1 or 2 diabetes. 

Clarify the terms like dyslipidemia, hba1c etc. when the authors use it in the statement.

What does the statement “the expression of the glial fibrillary acidic protein (GFAP), is increased” represents? Complete the statement.

Include a figure summarizing the molecular targets of acrolein in development of DR (not the potential molecules) that will the readers to comprehend the review.

Line 233; typo error ‘retia’

Author Response

The review is an elaborate compilation of related literature, however, as the title suggests, there are very limited study that presents a direct link of Acrolein in the development of diabetic retinopathy. Major portion of review is focused on either detailing about Acrolein induced toxicity or the role of biological factors involved in the pathogenesis of DR, but have only limited direct evidences. Author should remove extra text, which seems to be diverting the focus of the manuscript. Also, rather than just including a concluding statement of the referenced articles in the review, author should explain and discuss the statements.

Authors are extremely thankful to the reviewer for the constructive criticism and encouraging comments. We have addressed issues and revised the manuscript significantly with a more focused approach. We have modified the title, removed the unnecessary text, included the explanations and discussed the results of the referenced articles.

Minor comments

  1. Fig 1 is not necessary. It doesn’t seem to add anything in the manuscript.

Figure 1 is now removed in the revised manuscript.

  1. In the statement, “The positive correlation between diabetes and acrolein levels in diabetic patients possibly mediated through either lipid peroxidation or polyamines oxidation or a combination of both”, cell type is not clear. Explain.

This statement is modified and moved to the end of the paragraph to explain and discuss the cited observations. (Page no.4; lines 136-138)

  1. In the statement “Acrolein feeding increased the circulating levels of cholesterol and triglycerides within 24h in mice”, how acrolein involvement regulates cholesterol and triglycerides is not clear. Explain.

This statement is revised and explained in detail in the revised file. (Page no.4; lines 125-133)

  1. Contribution of Acrolein in type 1 and 2 diabetes is not distinguishable. Discuss with specifications in the review whether the study focused on type 1 or 2 diabetes. 

This is now discussed in the revised manuscript. (Page 3-4; lines 114-122)

  1. Clarify the terms like dyslipidemia, hba1c etc. when the authors use it in the statement.

These suggested changes are incorporated in the manuscript. Lines 119 and 132 (page 4).

  1. What does the statement “the expression of the glial fibrillary acidic protein (GFAP), is increased” represents? Complete the statement.

 We apologize for this error and is now corrected in the revised file. Page 4, lines 155-156.

  1. Include a figure summarizing the molecular targets of acrolein in development of DR (not the potential molecules) that will the readers to comprehend the review.

 This is an excellent suggestion and is included as new Fig 3.

  1. Line 233; typo error ‘retia’.

This error is fixed in the revised file

Reviewer 2 Report

In this review, Moaddey Alfarhan, Eissa Jafari and S. Priya Narayanan explains the role of acrolein as a mediator of oxidative damage in diabetic retinopathy with special focus on polyamine oxidation. This review is a valuable addition to the field as this review summarized the existing literature on the sources of acrolein, the impact of acrolein in the generation of oxidative damage in the diabetic retina, the mechanisms of acrolein action in the pathogenesis of DR, and therapeutic interventions to reduce acrolein toxicity.

General Comments:-

The manuscript is well written and figures are well presented. The article has interesting observations which is beneficial to researchers in the areas of acrolein, oxidative stress, retina and diabetic retinopathy associated complications.

Specific Comments:-

Page 2 Line 48 – role “in” DR progression.

Page 2 Line 83 – Acrolein, 4-HNE and MDA are the primary lipid aldehydes generated during lipid peroxidation. It will be also good to mention others in text.

Page 3 Line 98 – Please include more information about the identification and quantification of acrolein from published papers.

Page 4 Section 4.1 – Add information about Müller cells and its role in pathogenesis of DR, as its mentioned lot of times later in the review. It will be good to move information from Page 7 line 276 to here.

Page 5 Line 189 – elevates to elevated

Page 6 Line 260 – It will be good to have a separate section on Müller cell dysfunction caused by acrolein in both in vivo and in vitro. Please update figure 3 as well with Müller cell dysfunction

Page 6 Line 232 – retia to retina

Page 7 Section 5 – Please discuss more acrolein scavenging agents in this section

Author Response

In this review, Moaddey Alfarhan, Eissa Jafari and S. Priya Narayanan explains the role of acrolein as a mediator of oxidative damage in diabetic retinopathy with special focus on polyamine oxidation. This review is a valuable addition to the field as this review summarized the existing literature on the sources of acrolein, the impact of acrolein in the generation of oxidative damage in the diabetic retina, the mechanisms of acrolein action in the pathogenesis of DR, and therapeutic interventions to reduce acrolein toxicity.

General Comments:-

The manuscript is well written and figures are well presented. The article has interesting observations which is beneficial to researchers in the areas of acrolein, oxidative stress, retina and diabetic retinopathy associated complications.

Authors are grateful to the reviewer for the encouraging comments on our review article.

Specific Comments:

  1. Page 2 Line 48 – role “in” DR progression. This is corrected.

  1. Page 2 Line 83 – Acrolein, 4-HNE and MDA are the primary lipid aldehydes generated during lipid peroxidation. It will be also good to mention others in text.

This suggestion is included in the revised manuscript. (Page 2, lines 76-79)

  1. Page 3 Line 98 – Please include more information about the identification and quantification of acrolein from published papers.

This is a great suggestion and we have incorporated it in the revised file. (Page 3, lines 103-108)

  1. Page 4 Section 4.1 – Add information about Müller cells and its role in pathogenesis of DR, as its mentioned lot of times later in the review. It will be good to move information from Page 7 line 276 to here.

 This suggested change is included in the revised file.  Page 4, lines 151-156

     5. Page 5 Line 189 – elevates to elevated. This is corrected.

  1. Page 6 Line 260 – It will be good to have a separate section on Müller cell dysfunction caused by acrolein in both in vivo and in vitro. Please update figure 3 as well with Müller cell dysfunction.

These are excellent suggestions and we have now revised file accordingly. Muller cell dysfunction is described as a separate section (section 5.4, page 8, lines 304-316). Modified figure 3 is Figure 2 in the revised submission.

  1. Page 6 Line 232 – retia to retina. This error is fixed in the revised file.

 Page 7 Section 5 – Please discuss more acrolein scavenging agents in this section.

This section is elaborated with more information on acrolein scavenging agents. Page 9, lines 328-332, 339-341, and 354-361

Reviewer 3 Report

Although the topic of this work is quite interesting and of scientific relevance, the manuscript is quite confusing, with the concepts needing to be revised, using more appropriate references. A better structure should be defined to present the subtopics reviewed by the authors. Importantly syntax and grammar must be reviewed, as large parts of the manuscript are confusing and difficult to understand.

Author Response

Although the topic of this work is quite interesting and of scientific relevance, the manuscript is quite confusing, with the concepts needing to be revised, using more appropriate references. A better structure should be defined to present the subtopics reviewed by the authors. Importantly syntax and grammar must be reviewed, as large parts of the manuscript are confusing and difficult to understand.

Authors are thankful for the constructive criticism provided by the reviewer. We have modified the structure and reorganized the subtopics. In the revised file, we have included all the relevant literature on diabetic retinopathy and acrolein. We have revised the file to correct syntax and grammatical errors.